# Lower and Upper Bounds on the Pseudo-Dimension of Tensor Network Models

**Behnoush Khavari**
DIRO & Mila
Université de Montréal
behnoush.khavari@umontreal.ca

**Guillaume Rabusseau**
DIRO & Mila, CIFAR AI Chair
Université de Montréal
grabus@iro.umontreal.ca

## Abstract

Tensor network (TN) methods have been a key ingredient of advances in condensed matter physics and have recently sparked interest in the machine learning community for their ability to compactly represent very high-dimensional objects. TN methods can for example be used to efficiently learn linear models in exponentially large feature spaces [56]. In this work, we derive upper and lower bounds on the VC-dimension and pseudo-dimension of a large class of TN models for classification, regression and completion. Our upper bounds hold for linear models parameterized by arbitrary TN structures, and we derive lower bounds for common tensor decomposition models (CP, Tensor Train, Tensor Ring and Tucker) showing the tightness of our general upper bound. These results are used to derive a generalization bound which can be applied to classification with low-rank matrices as well as linear classifiers based on any of the commonly used tensor decomposition models. As a corollary of our results, we obtain a bound on the VC-dimension of the matrix product state classifier introduced in [56] as a function of the so-called bond dimension (i.e. tensor train rank), which answers an open problem listed by Cirac, Garre-Rubio and Pérez-García in [13].

## 1   Introduction

Tensor networks (TNs) have emerged in the quantum physics community as a mean to compactly represent wave functions of large quantum systems [45, 5, 52]. Their introduction in physics can be traced back to the work of Penrose [47] and Feynman [15]. Akin to matrix factorization, TN methods rely on factorizing a high-order tensor into small factors and have recently gained interest from the machine learning community for their ability to efficiently represent and perform operations on very high-dimensional data and high-order tensors. They have been for example successfully used for compressing models [43, 69, 42, 29, 70], developing new insights on the expressiveness of deep neural networks [14, 31] and designing novel approaches to supervised [56, 18] and unsupervised [55, 25, 39] learning. Most of these approaches leverage the fact that TN can be used to efficiently parameterize high-dimensional linear maps, which is appealing from two perspectives: it makes it possible to learn models in exponentially large feature spaces *and* it acts as a regularizer, controlling the capacity of the class of hypotheses considered for learning.

While the expressive power of TN models has been studied recently [17, 2], the focus has mainly been on the representation capacity of TN models, but not on their ability to *generalize* in the context of supervised learning tasks. In this work, we study the generalization ability of TN models by deriving lower and upper bounds on the VC-dimension and pseudo-dimension of TN models commonly used for classification, completion and regression, from which bounds on the generalization gap of TN models can be derived. Using the general framework of tensor networks, we derive a general upper bound for models parameterized by *arbitrary* TN structures, which applies to all commonly used tensor decomposition models [20] such as CP [27], Tucker [59] and tensor train (TT)) [46], as well

35th Conference on Neural Information Processing Systems (NeurIPS 2021).

as more sophisticated structures including hierarchical Tucker [19, 23], tensor ring (TR) [73] and projected entangled state pairs (PEPS) [60].

Our analysis proceeds mainly in two steps. First, we formally define the notion of TN learning model by disentangling the underlying graph structure of a TN from its parameters (the core tensors, or factors, involved in the decomposition). This allows us to define, in a conceptually simple way, the hypothesis class $\mathcal{H}_G$ corresponding to the family of linear models whose weights are represented using an arbitrary TN structure $G$. We then proceed to deriving upper bounds on the VC/pseudo-dimension and generalization error of the class $\mathcal{H}_G$. These bounds follow from a classical result from Warren [66] which was previously used to obtain generalization bounds for neural networks [3], matrix completion [54] and tensor completion [41]. The bounds we derive naturally relate the capacity of $\mathcal{H}_G$ to the underlying graph structure $G$ through the number of nodes and effective number of parameters of the TN. To assess the tightness of our general upper bound, we derive lower bounds for particular TN structures (rank-one, CP, Tucker, TT and TR). These lower bounds show that, for completion, regression and classification, our general upper bound is tight up to a log factor for rank-one, TT and TR tensors, and is tight up to a constant for matrices. Lastly, as a corollary of our results, we obtain a bound on the VC-dimension of the tensor train classifier introduced in [56], which answers one of the open problems listed by Cirac, Garre-Rubio and Pérez-García in [13].

**Related work**  Machine learning models using low-rank parametrization of the weights have been investigated (mainly from a practical perspective) for various decomposition models, including low-rank matrices [36, 49, 67], CP [1, 37, 7], Tucker [35, 16, 26, 50], tensor train [48, 10, 44, 56, 18, 53, 11, 65, 68] and PEPS [12]. From a more theoretical perspective, generalization bounds for matrix and tensor completion have been derived in [54, 41] (based on the Tucker format for the tensor case). A bound on the VC-dimension of low-rank matrix classifiers was derived in [67] and a bound on the pseudo-dimension of regression functions whose weights have low Tucker rank was given in [50] (for both these cases, we show that our results improve over these previous bounds, see Section 4.2). To the best of our knowledge the VC-dimension of tensor train classifiers has not been studied in the past, but the statistical consistency of the convex relaxation of the tensor completion problem was studied in [58, 57] for the Tucker decomposition and in [28] for the tensor train decomposition. Lastly, in [38] the authors study the complexity of learning with tree tensor networks using the notion of metric entropy and covering numbers. They provide generalization bounds which are qualitatively similar to ours, but their results only hold for TN structures whose underlying graph is a tree (thus excluding models such as CP, tensor ring and PEPS) and they do not provide lower bounds.

**Summary of contributions**  We introduce a *unifying framework for TN-based learning models*, which generalizes a wide range of models based on tensor factorization for completion, classification and regression. This framework allows us to consider the class $\mathcal{H}_G$ of low-rank TN models for a given *arbitrary TN structure $G$* (Section 3). We provide general *upper bounds on the pseudo-dimension and VC-dimension* of the hypothesis class $\mathcal{H}_G$ *for arbitrary TN structure $G$* for regression, classification and completion. Our results naturally relate the capacity of $\mathcal{H}_G$ to the number of parameters of the underlying TN structure $G$ (Section 4.1). From these results, we derive a *generalization bound for TN-based classifiers parameterized by arbitrary TN structures* (Theorem 4). We compare our results to previous bounds for specific decomposition models and show that our general upper bound is always of the same order and sometimes even improves on previous bounds (Section 4.2). We derive several lower bounds showing that our general upper bound is tight up to a log factor for particular TN structures (Section 5). A summary of the lower bounds derived in this work, as well as upper bounds implied by our general result for particular TN structures, can be found in Table 1 at the end of the paper.

## 2   Preliminaries

In this section, we present basic notions of tensor algebra and tensor networks as well as generalization bounds based on combinatorial complexity measures. We start by introducing some notations. For any integer $k$ we use $[k]$ to denote the set of integers from 1 to $k$. We use lower case bold letters for vectors (e.g. $\mathbf{v} \in \mathbb{R}^{d_1}$), upper case bold letters for matrices (e.g. $\mathbf{M} \in \mathbb{R}^{d_1 \times d_2}$) and bold calligraphic letters for higher order tensors (e.g. $\boldsymbol{\mathcal{T}} \in \mathbb{R}^{d_1 \times d_2 \times d_3}$). The inner product of two $k$-th order tensors $\boldsymbol{\mathcal{S}}, \boldsymbol{\mathcal{T}} \in \mathbb{R}^{d_1 \times \cdots \times d_k}$ is defined by $\langle \boldsymbol{\mathcal{T}}, \boldsymbol{\mathcal{S}} \rangle = \sum_{i_1=1}^{d_1} \cdots \sum_{i_k=1}^{d_k} \boldsymbol{\mathcal{T}}_{i_1 \ldots i_k} \boldsymbol{\mathcal{S}}_{i_1 \ldots i_k}$. The outer product of

Figure 1: Tensor network representation of common operations on matrices and vectors.

two vectors $\mathbf{u} \in \mathbb{R}^{d_1}$ and $\mathbf{v} \in \mathbb{R}^{d_2}$ is denoted by $\mathbf{u} \otimes \mathbf{v} \in \mathbb{R}^{d_1 \times d_2}$ with elements $(\mathbf{u} \otimes \mathbf{v})_{i,j} = \mathbf{u}_i \mathbf{v}_j$. The outer product generalizes to an arbitrary number of vectors. We use the notation $(\mathbb{R}^d)^{\otimes p}$ to denote the space of $p$-th order hypercubic tensors of size $d \times d \times \cdots \times d$. We denote by $\mathcal{Y}^{\mathcal{X}}$ the space of functions $f : \mathcal{X} \mapsto \mathcal{Y}$. $\text{sign}(\cdot)$ stands for the sign function. Finally, given a graph $G = (V, E)$ and a vertex $v \in V$, we denote by $E_v = \{e \in E \mid v \in e\}$ the set of edges incident to the vertex $v$.

## 2.1 Tensors and Tensor Networks

**Tensor networks** A *tensor* $\mathcal{T} \in \mathbb{R}^{d_1 \times \cdots \times d_p}$ can simply be seen as a multidimensional array $(\mathcal{T}_{i_1, \cdots, i_p} : i_n \in [d_n], n \in [p])$. Complex operations on tensors can be intuitively represented using the graphical notation of tensor network (TN) diagrams [5, 45]. In tensor networks, a $p$-th order tensor is illustrated as a node with $p$ edges (or *legs*) in a graph  . An edge between two nodes of a TN represents a contraction over the corresponding modes of the two tensors. Consider the following simple TN with two nodes:  . The first node represents a matrix $\mathbf{A} \in \mathbb{R}^{m \times n}$ and the second one a vector $\mathbf{x} \in \mathbb{R}^n$. Since this TN has one dangling leg (i.e. an edge which is not connected to any other node), it represents a first order tensor, i.e. a vector. The edge between the second leg of $\mathbf{A}$ and the leg of $\mathbf{x}$ corresponds to a contraction between the second mode of $\mathbf{A}$ and the first mode of $\mathbf{x}$. Hence, the resulting TN represents the classical matrix-product, which can be seen by calculating the $i$-th component of this TN:  $= \sum_j \mathbf{A}_{ij} \mathbf{x}_j = (\mathbf{A}\mathbf{x})_i$ . Other examples of TN representations of common operations on matrices and vectors can be found in Figure 1. A special case of TN is the tensor train decomposition [46] which factorizes a $n$-th order tensor $\mathcal{T}$ in the form  . This corresponds to

$$\mathcal{T}_{i_1, i_2, \ldots, i_n} = \sum_{\alpha_1=1}^{r_1} \cdots \sum_{\alpha_{n-1}=1}^{r_{n-1}} (\mathcal{G}_1)_{i_1, \alpha_1} (\mathcal{G}_2)_{\alpha_1, i_2, \alpha_2} \ldots (\mathcal{G}_{n-1})_{\alpha_{n-2}, i_{n-1}, \alpha_{n-1}} (\mathcal{G}_n)_{\alpha_{n-1}, i_n} \quad (1)$$

where the tuple $(r_i)_{i=1}^{n-1}$ associated with the TT representation is called TT-rank.

**Tensor network structures** A tensor network (TN) can be fundamentally decomposed in two constituent parts: a tensor network structure, which describes its graphical structure, and a set of core tensors assigned to each node. For example, the tensor in $\mathbb{R}^{d_1 \times d_2 \times d_3 \times d_4}$ represented by the TN  is obtained by assigning the core tensors $\mathcal{T} \in \mathbb{R}^{d_1 \times d_2 \times R}$ and $\mathcal{S} \in \mathbb{R}^{R \times d_3 \times d_4}$ to the nodes of the TN structure  .

Formally, a *tensor network structure* is given by a graph $G = (V, E, dim)$ where edges are labeled by integers: $V$ is the set of vertices, $E \subset V \cup (V \times V)$ is a set of edges containing both classical edges ($e \in V \times V$) and singleton edges ($e \in V$) and $dim : E \to \mathbb{N}$ assigns a dimension to each edge in the graph. The set of singleton edges $\delta_G = E \cap V$ corresponds to the dangling legs of a TN. Given a TN structure $G$, one obtains a tensor by assigning a core tensor $\mathcal{T}^v \in \bigotimes_{e \in E_v} \mathbb{R}^{dim(e)}$ to each vertex $v$ in the graph, where $E_v = \{e \in E \mid v \in e\}$. The resulting tensor, denoted by $TN(G, \{\mathcal{T}^v\}_{v \in V})$, is a tensor of order $|\delta_G|$ in the tensor product space $\bigotimes_{e \in \delta_G} \mathbb{R}^{dim(e)}$. Given a tensor structure $G = (V, E, dim)$, the set of all tensors that can be obtained by assigning core tensors to the vertices of $G$ is denoted by $\mathcal{T}(G) \subset \bigotimes_{e \in \delta_G} \mathbb{R}^{dim(e)}$:

$$\mathcal{T}(G) = \{TN(G, \{\mathcal{T}^v\}_{v \in V}) : \mathcal{T}^v \in \bigotimes_{e \in E_v} \mathbb{R}^{dim(e)}, v \in V\}. \quad (2)$$

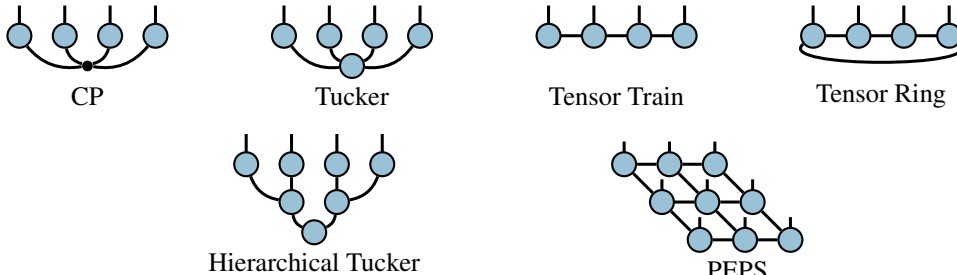

Figure 2: TN representation of common decomposition models for 4th order and 9th order tensors. For CP, the black dot represents a hyperedge corresponding to a joint contraction over 4 indices.

As an illustration, one can check that the set of $m \times n$ matrices of rank at most $r$ is equal to $\mathcal{T}(\,{}^m\!\!-\!\!\bigcirc\!\!-\!\!{}^r\!\!-\!\!\bigcirc\!\!-\!\!{}^n\,)$. Similarly, the set of 4th order $d$-dimensional tensors of TT rank at most $r$ is equal to $\mathcal{T}(\,\bigcirc\!\!-\!\!{}^r\!\!-\!\!\bigcirc\!\!-\!\!{}^r\!\!-\!\!\bigcirc\!\!-\!\!{}^r\!\!-\!\!\bigcirc\,)$.

Finally, for a given graph structure $G$, the number of parameters of any member of the family $\mathcal{T}(G)$ in Equation (2) (which is the total number of entries of the core tensors $\{\mathcal{T}^v\}_{v \in V}$) is given by

$$N_G = \sum_{v \in V} \prod_{e \in E_v} dim(e) \tag{3}$$

This will be a central quantity in the generalization bounds and bounds on the VC-dimension of TN models we derive in Section 4.

**Common tensor network structures**    In Figure 2, we show the tensor network structures associated with classical tensor decomposition models such as CP, Tucker [59] and tensor train (TT) [46], also known as matrix product state (MPS) [45, 52]. For the case of the Candecomp/Parafac (CP) decomposition [27], note that the TN structure is a hyper-graph rather than a graph. We introduced the notion of TN structure focusing on graphs for clarity of exposition in the previous paragraph, but our formalism and results can be straightforwardly extended to hyper-graph TN structures. In addition, we include the tensor ring (TR) [73] (also known as periodic MPS) and PEPS decompositions which have initially emerged in quantum physics and recently gained interest in the machine learning community (see e.g., [12, 62, 63, 71]). We also show the hierarchical Tucker decomposition initially introduced in [19, 23].

## 2.2 Generalization Bound and Complexity Measures

The goal of supervised learning is to learn a function $f$ mapping inputs $x \in \mathcal{X}$ to outputs $y \in \mathcal{Y}$ from a sample of input-output examples $S = \{(x_1, y_1), \cdots, (x_n, y_n)\}$ drawn independently and identically (i.i.d.) from an unknown distribution $D$, where each $y_i \simeq f(x_i)$. Given a set of hypotheses $\mathcal{H} \subset \mathcal{Y}^{\mathcal{X}}$, one natural objective is to find the hypothesis $h \in \mathcal{H}$ minimizing the *risk* $R(h) = \mathbb{E}_{(x,y) \sim D} \ell(h(x), y)$ where $\ell : \mathcal{Y} \times \mathcal{Y} \to \mathbb{R}_+$ is a loss function measuring the quality of the predictions made by $h$. However, since the distribution $D$ is unknown, machine learning algorithms often rely on the *empirical risk minimization* principle which consists in finding the hypothesis $h \in \mathcal{H}$ that minimizes the *empirical risk* $\hat{R}_S(h) = \frac{1}{n} \sum_{i=1}^{n} \ell(h(x_i), y_i)$. It is easy to see that the empirical risk is an unbiased estimator of the risk and one of the concerns of learning theory is to provide guarantees on the quality of this estimator. Such guarantees include *generalization bounds*, which are probabilistic bounds on the *generalization gap* $R(h) - \hat{R}_S(h)$. The generalization gap naturally depends on the size of the sample $S$, but also on the richness (or *capacity*, complexity) of the hypothesis class $\mathcal{H}$.

In this work, our focus is on *uniform* generalization bounds, which bound the generalization gap uniformly for any hypothesis $h \in \mathcal{H}$ as a function of the size of the training sample and of the complexity of the hypothesis class $\mathcal{H}$. While there are many ways of measuring the complexity of $\mathcal{H}$, including VC-dimension, Rademacher complexity, metric entropy and covering numbers, we focus on the *VC-dimension* for classification tasks and its counterpart for real-valued functions, the *pseudo-dimension*, for completion and regression tasks.

**Definition 1.** *Let $\mathcal{H} \subset \{-1, +1\}^{\mathcal{X}}$ be a hypothesis class. The* growth function $\Pi_{\mathcal{H}} : \mathbb{N} \to \mathbb{N}$ *of $\mathcal{H}$ is defined by*

$$\Pi_{\mathcal{H}}(n) = \sup_{S = \{x_1, \ldots, x_n\} \subset \mathcal{X}} |\{(h(x_1), \ldots, h(x_n)) \mid h \in \mathcal{H}\}|.$$

*The* VC-dimension *of $\mathcal{H}$, $d_{\mathrm{VC}}(\mathcal{H})$, is the largest number of points $x_1, \cdots, x_n$ shattered by $\mathcal{H}$, i.e., for which $|\{(h(x_1), \ldots, h(x_n)) \mid h \in \mathcal{H}\}| = 2^n$. In other words: $d_{\mathrm{VC}}(\mathcal{H}) = \sup\{n \mid \Pi_{\mathcal{H}}(n) = 2^n\}$.*

*For a real-valued hypothesis class $\mathcal{H} \subset \mathbb{R}^{\mathcal{X}}$, we say that $\mathcal{H}$ pseudo-shatters the points $x_1, \ldots, x_n \in \mathcal{X}$ with thresholds $t_1, \ldots, t_n \in \mathbb{R}$, if for every binary labeling of the points $(s_1, \ldots, s_n) \in \{-1, +1\}^n$, there exists $h \in \mathcal{H}$ s.t. $h(x_i) < t_i$ if and only if $s_i = -1$.*

*The* pseudo-dimension *of a real-valued hypothesis class $\mathcal{H} \subset \mathbb{R}^{\mathcal{X}}$, $\mathrm{Pdim}(\mathcal{H})$, is the supremum over $n$ for which there exist $n$ points that are pseudo-shattered by $\mathcal{H}$ (with some thresholds).*

Pseudo-dimension and VC-dimension are combinatorial measures of complexity (or capacity) which can be used to derive classical uniform generalization bounds over a hypothesis class (see, e.g., [6, 40, 3]). By definition, the pseudo-dimension is related to the notion of VC-dimension by the relation

$$\mathrm{Pdim}(\mathcal{H}) = d_{\mathrm{VC}}(\{(x, t) \mapsto \mathrm{sign}(h(x) - t) \mid h \in \mathcal{H}\})$$

which holds for any $\mathcal{H} \subset \mathbb{R}^{\mathcal{X}}$.

## 3  Tensor Networks for Supervised Learning

In this section, we formalize the general notion of *tensor network models*. We then show how it encompasses classical models such as low-rank matrix completion [8, 9, 22, 51], classification [36, 49, 67], and tensor train based models [56, 18, 53, 11, 65, 68].

### 3.1  Tensor Network Learning Models

Consider a classification problem where the input space $\mathcal{X}$ is the space of $p$-th order tensors $\mathbb{R}^{d_1 \times d_2 \times \cdots \times d_p}$. One motivation for TN models is that the tensor product space $\mathcal{X}$ can be exponentially large, thus learning a linear model in this space is often not feasible. Indeed, the number of parameters of a linear classifier $h : \mathcal{X} \mapsto \mathrm{sign}(\langle \mathcal{X}, \mathcal{W} \rangle)$, where $\mathcal{W} \in \mathbb{R}^{d_1 \times \cdots \times d_p}$ is the tensor weight parameters, grows exponentially with $p$. TN models parameterize $\mathcal{W}$ as a low-rank TN, thus reducing the number of parameters needed to represent a model $h$. Our objective is to derive generalization bounds for the class of such hypotheses parameterized by low-rank tensor networks for classification, regression and completion tasks.

Formally, let $G = (V, E, \dim)$ be a TN structure for tensors of shape $d_1 \times \cdots \times d_p$, i.e. where the set of singleton edges $\delta_G = E \cap V = \{v_1, \cdots, v_p\}$ and $\dim(v_i) = d_i$ for each $i \in [p]$. We are interested in the class of models whose weight tensors are represented in the TN structure $G$:

$$\mathcal{H}_G^{\mathrm{regression}} = \{h : \mathcal{X} \mapsto \langle \mathcal{W}, \mathcal{X} \rangle \mid \mathcal{W} \in \mathcal{T}(G)\} \tag{4}$$

$$\mathcal{H}_G^{\mathrm{classif}} = \{h : \mathcal{X} \mapsto \mathrm{sign}(\langle \mathcal{W}, \mathcal{X} \rangle) \mid \mathcal{W} \in \mathcal{T}(G)\} \tag{5}$$

$$\mathcal{H}_G^{\mathrm{completion}} = \{h : (i_1, \cdots, i_p) \mapsto \mathcal{W}_{i_1, \cdots, i_p} \mid \mathcal{W} \in \mathcal{T}(G)\} \tag{6}$$

In Equation (6) for the completion hypothesis class, $p$-th order tensors are interpreted as real-valued functions $f : [d_1] \times \cdots \times [d_p] \mapsto \mathbb{R}$ over the indices of the tensor. $\mathcal{H}_G^{\mathrm{completion}}$ is thus a class of functions over the indices domain, for which the notion of pseudo-dimension is well-defined. This treatment of completion as a supervised learning task was considered previously to derive generalization bounds for matrix and tensor completion [54, 41].

The benefit of TN models comes from the drastic reduction in parameters when the TN structure $G$ is *low-rank*, in the sense that the number of parameters $N_G$ is small compared to $d_1 d_2 \cdots d_p$. In addition to allowing one to represent linear models in exponentially large spaces, this compression controls the capacity of the corresponding hypothesis class $\mathcal{H}_G$.

### 3.2  Examples

To illustrate some TN models, we now present several examples of models based on common TN structures: low-rank matrices and tensor trains.

**Low-rank matrices**   As discussed in Section 2.1, if we define the TN structure

$$G_{\mathrm{mat}}(r) = \overset{d_1}{\underset{}{\bigcirc}}\!\!\overset{r}{\rule{0pt}{0pt}}\!\!\overset{d_2}{\bigcirc}\ ,$$

then $\mathcal{T}(G_{\mathrm{mat}}(r))$ is the set of matrices in $\mathbb{R}^{d_1 \times d_2}$ of rank at most $r$. The hypothesis class $\mathcal{H}^{\mathrm{completion}}_{G_{\mathrm{mat}}(r)}$ then corresponds to the classical problem of low-rank matrix completion [8, 9, 22, 51]. Similarly $\mathcal{H}^{\mathrm{classif}}_{G_{\mathrm{mat}}(r)}$ corresponds to the hypothesis class of low-rank matrix classifiers. This hypothesis class was previously considered, notably to compactly represent the parameters of support vector machines for matrix inputs [36, 49, 67]. Lastly, for the regression case, $\mathcal{H}^{\mathrm{regression}}_{G_{\mathrm{mat}}(r)}$ is the set of functions $\{h : \mathbf{X} \mapsto \mathrm{Tr}(\mathbf{W}\mathbf{X}^\top) \mid \mathrm{rank}(\mathbf{W}) \leq r\}$. Learning hypotheses from this class is relevant in, e.g., quantum tomography, where it is known as the *low-rank trace regression* problem [24, 64, 30, 33].

**Tensor train tensors**   The tensor train (TT) decomposition model [46] also known as matrix product state (MPS) in the quantum physics community [45, 52], has a number of parameters that grows only linearly with the order of the tensor. This makes the TT format an appealing model for compressing the parameters of ML models [56, 44, 17, 43]. We now present the tensor train classifier model which was introduced in [56] and subsequently explored in [18]. Given a vector input $\mathbf{x} \in \mathbb{R}^p$, Stoudenmire and Schwab [56] propose to map $\mathbf{x}$ into a high-dimensional space of $p$-th order tensors $\mathcal{X} = \mathbb{R}^{d \times \cdots \times d}$ by applying a local feature map $\phi : \mathbb{R} \to \mathbb{R}^d$ to each component of the vector $\mathbf{x}$ and taking their outer product: $\Phi(\mathbf{x}) = \phi(\mathbf{x}_1) \otimes \phi(\mathbf{x}_2) \otimes \cdots \otimes \phi(\mathbf{x}_p) \in (\mathbb{R}^d)^{\otimes p}$.

Instead of relying on the so-called kernel trick, Stoudenmire and Schwab propose to directly learn the parameters $\mathcal{W}$ of a linear model $h : \mathbf{x} \mapsto \mathrm{sign}(\langle \mathcal{W}, \Phi(\mathbf{x})\rangle)$ in the exponentially large feature space $\mathcal{X}$. The learning problem is made tractable by paremeterizing $\mathcal{W}$ as a low-rank TT tensor (see Equation (1)). Letting

$$G_{\mathrm{TT}}(r_1, \cdots, r_{p-1}) = \underset{d_1}{\bigcirc}\!\overset{r_1}{\rule{0pt}{0pt}}\!\underset{d_2}{\bigcirc}\!\overset{r_2}{\rule{0pt}{0pt}}\cdots\overset{r_{p-2}}{\rule{0pt}{0pt}}\!\underset{d_{p-1}}{\bigcirc}\!\overset{r_{p-1}}{\rule{0pt}{0pt}}\!\underset{d_p}{\bigcirc}$$

the hypothesis class considered in [56] is $\mathcal{H}^{\mathrm{classif}}_{G_{\mathrm{TT}}(r_1, \cdots, r_{p-1})}$. In addition to the approach of [56], which was extended in [18] and [53], tensor train classifiers were also previously considered in [11, 65, 68]. Similarly, the hypothesis class $\mathcal{H}^{\mathrm{completion}}_{G_{\mathrm{TT}}(r_1, \cdots, r_{p-1})}$ corresponds to the low-rank TT completion problem [21, 48, 61].

**Other TN models**   Lastly, we mention that our formalism can be applied to any tensor models having a low-rank structure, including CP, Tucker, tensor ring and PEPS. As mentioned previously, for the case of the CP decomposition, the graph $G$ of the TN structure is in fact a hyper-graph with $|V| = p$ nodes and $N_G = pdr$ parameters for a weight tensor in $(\mathbb{R}^d)^{\otimes p}$ with CP rank at most $r$. Several TN learning models using these decomposition models have been proposed previously, including [26, 50] for regression in the Tucker format, [12] for classification using the PEPS model, [37, 7] for classification with the CP decomposition and [62, 72] for tensor completion with TR.

# 4   Pseudo-dimension and Generalization Bounds for Tensor Network Models

In this section, we give a general upper bound on the VC-dimension and pseudo-dimension of hypothesis classes parameterized by *arbitrary* TN structures for regression, classification and completion. We then discuss corollaries of this general upper bound for common TN models including low-rank matrices and TT tensors, and compare them with existing results. Examples of particular upper bounds that can be derived from our general result can be found in Table 1.

## 4.1   Upper Bounds on the VC-dimension, Pseudo-dimension and Generalization Gap

The following theorem states one of our main results which upper bounds the VC and pseudo-dimension of models parameterized by arbitrary TN structures.

**Theorem 2.** *Let $G = (V, E, \dim)$ be a tensor network structure and let $\mathcal{H}^{regression}_G$, $\mathcal{H}^{classif}_G$, $\mathcal{H}^{completion}_G$ be the corresponding hypothesis classes defined in Equations (4-6), where each model has $N_G$ parameters (see Equation (3)).*

*Then, $\mathrm{Pdim}(\mathcal{H}^{regression}_G)$, $d_{\mathrm{VC}}(\mathcal{H}^{classif}_G)$ and $\mathrm{Pdim}(\mathcal{H}^{completion}_G)$ are all upper bounded by $2N_G \log(12|V|)$.*

These bounds naturally relate the capacity of the TN classes $\mathcal{H}_G^{\text{regression}}, \mathcal{H}_G^{\text{classif}}, \mathcal{H}_G^{\text{completion}}$ to the number of parameters $N_G$ of the underlying TN structure $G$. Following the analysis of [54] for matrix completion and its extension to the Tucker decomposition model presented in [41], the proof of this theorem leverages Warren's theorem which bounds the number of sign patterns a system of polynomial equations can take.

**Theorem 3** ([66]). *The number of sign patterns of $n$ real polynomials, each of degree at most $v$, over $N$ variables is at most $\left(\frac{4evn}{N}\right)^N$ for all $n > N > 2$ (where $e$ is Euler's number).*

The proof of Theorem 2 fundamentally relies on Warren's theorem to bound the number of sign patterns that can be achieved by hypotheses in $\mathcal{H}_G^{\text{regression}}$ on a set of $n$ input examples $\boldsymbol{\mathcal{X}}_1, \cdots, \boldsymbol{\mathcal{X}}_n$. Indeed, the set of predictions $y_i = h(\boldsymbol{\mathcal{X}}_i)$ for $i \in [n]$ realizable by hypotheses $h \in \mathcal{H}_G^{\text{regression}}$ can be seen as a set of $n$ polynomials of degree $|V|$ over $N_G$ variables. The variables of the polynomials are the entries of the core tensors $\{\boldsymbol{\mathcal{T}}^v\}_{v \in V}$. The upper bound on the number of sign patterns obtained from Warren's theorem can then be leveraged to obtain a bound on the pseudo-dimension of the hypothesis class $\mathcal{H}_G^{\text{regression}}$, which in turn implies the upper bounds on $d_{\text{VC}}(\mathcal{H}_G^{\text{classif}})$ and $\text{Pdim}(\mathcal{H}_G^{\text{completion}})$. The complete proof of Theorem 2 can be found in Appendix A.1.1.

Note that Theorem 2 implies that for a fixed number of parameters $N_G$, the VC-dimension grows with the number of vertices in the TN, thus a higher-order tensorization increases the capacity as measured by the VC dimension. This supports a common observation that higher-order tensorizations of high-dimensional data generally result in a model with better learning capacity.

The bounds on the VC-dimension and pseudo-dimension presented in Theorem 2 can be leveraged to derive bounds on the generalization error of the corresponding learning models; see for example [40]. In the following theorem, we derive such a generalization bound for classifiers parameterized by arbitrary TN structures.

**Theorem 4.** *Let $S$ be a sample of size $n$ drawn from a distribution $D$ and let $\ell$ be a loss bounded by 1. Then, for any $\delta > 0$, with probability at least $1 - \delta$ over the choice of $S$, for any $h \in \mathcal{H}_G^{\text{classif}}$,*

$$R(h) < \hat{R}_S(h) + 2\sqrt{\frac{2}{n}\left(N_G \log \frac{8en|V|}{N_G} + \log \frac{4}{\delta}\right)}. \tag{7}$$

The proof of this theorem, which can be found in Appendix A.1.2, relies on a symmetrization lemma and a corollary of Hoeffding's inequality. It follows from this theorem that, with high probability, the generalization gap $R(h) - \hat{R}_S(h)$ of any hypothesis $h \in \mathcal{H}_G^{\text{classif}}$ is in $\mathcal{O}\left(\sqrt{\frac{N_G \log(n)}{n}}\right)$. This bound naturally relates the sample complexity of the hypothesis class with its expressiveness. The notion of richness of the hypothesis class appearing in this bound reflects the structure of the underlying TN through the number of parameters $N_G$. Using classical results (see, e.g., Theorem 10.6 in [40]), similar generalization bounds for regression and classification with arbitrary TN structures can be obtained from the bounds on the pseudo-dimension of $\mathcal{H}_G^{\text{regression}}$ and $\mathcal{H}_G^{\text{completion}}$ derived in Theorem 2. To examine this upper bound in practice, we perform an experiment with low-rank TT classifiers on synthetic data which can be found in Appendix B.

In the next subsection, we present corollaries of our results for particular TN structures, including low-rank matrix completion and the TT classifiers introduced in [56].

## 4.2 Special cases

We now discuss special cases of Theorems 2 and 4 and compare them with existing results.

**Low-rank matrices**    Let $G_{\text{mat}}(r) = \overset{d_1}{\bigcirc} \overset{r}{\text{—}} \overset{d_2}{\bigcirc}$ and $\mathcal{T}(G_{\text{mat}}(r))$ be the set of $d_1 \times d_2$ matrices of rank at most $r$. In this case, we have $|V| = 2$ and $N_{G_{\text{mat}(r)}} = r(d_1 + d_2)$, and Theorems 2 and 4 give the following result.

**Corollary 5.** $\text{Pdim}(\mathcal{H}_{G_{\text{mat}(r)}}^{\text{regression}})$, $d_{\text{VC}}(\mathcal{H}_{G_{\text{mat}(r)}}^{\text{classif}})$ *and* $\text{Pdim}(\mathcal{H}_{G_{\text{mat}(r)}}^{\text{completion}})$ *are all upper bounded by* $10r(d_1 + d_2)$. *Moreover, with high probability over the choice of a sample $S$ of size $n$ drawn i.i.d.*

*from a distribution $D$, the generalization gap $R(h) - \hat{R}_S(h)$ of any hypothesis $h \in \mathcal{H}^{classif}_{G_{\mathrm{mat}}(r)}$ is in $\mathcal{O}\left(\sqrt{\frac{r(d_1+d_2)\log(n)}{n}}\right)$.*

This bound improves on the one given in [67] where the VC-dimension of $\mathcal{H}^{\mathrm{classif}}_{G_{\mathrm{mat}}(r)}$ is bounded by $r(d_1+d_2)\log(r(d_1+d_2))$ (see Theorem 2 in [67]). For the matrix completion case, our upper bound improves on the bound $\mathrm{Pdim}(\mathcal{H}^{\mathrm{completion}}_{G_{\mathrm{mat}}(r)}) \leq r(d_1+d_2)\log\frac{16ed_1}{r}$ derived in [54]. In Section 5, we will derive lower bounds showing that the upper bounds on the VC/pseudo-dimension of Corollary 5 are tight up to the constant factor 10 for matrix completion, regression and classification.

**Tensor train**   Let $G_{\mathrm{TT}}(r) = $  and $\mathcal{T}(G_{\mathrm{TT}}(r))$ be the set of tensors of TT rank at most $r$. In this case, we have $|V| = p$ and $N_G = \mathcal{O}\left(dpr^2\right)$ where $d = \max_i d_i$. For this class of hypotheses, Theorems 2 and 4 give the following result.

**Corollary 6.** $\mathrm{Pdim}(\mathcal{H}^{regression}_{G_{\mathrm{TT}}(r)})$, $d_{\mathrm{VC}}(\mathcal{H}^{classif}_{G_{\mathrm{TT}}(r)})$ and $\mathrm{Pdim}(\mathcal{H}^{completion}_{G_{\mathrm{TT}}(r)})$ are all in $\mathcal{O}\left(dpr^2\log(p)\right)$, where $d = \max_i d_i$. Moreover, with high probability over the choice of a sample $S$ of size $n$ drawn i.i.d. from a distribution $D$, the generalization gap $R(h) - \hat{R}_S(h)$ of any hypothesis $h \in \mathcal{H}^{classif}_{G_{\mathrm{TT}}(r)}$ is in $\mathcal{O}\left(\sqrt{\frac{dpr^2\log(n)}{n}}\right)$.

This result applies for the MPS model introduced in [56] and thus answers the open problem listed as Question 13 in [13]. To the best of our knowledge, the VC-dimension of tensor train classifier models has not been studied previously and our work is the first to address this open question. The lower bounds we derive in Section 5 show that the upper bounds on the VC/pseudo-dimension of Corollary 6 are tight up to a $\mathcal{O}\left(\log(p)\right)$ factor.

**Tucker**   We briefly compare our result with the ones proved in [41] for tensor completion and in [50] for tensor regression using the Tucker decomposition. For a Tucker decomposition with maximum rank $r$ for tensors of size $d_1 \times \cdots \times d_p$ with maximal dimension $d = \max_i d_i$, the number of parameters is in $\mathcal{O}\left(r^p + dpr\right)$ and the number of vertices in the TN structure is $p + 1$. In this case, Theorems 2 and 4 show that the VC/pseudo-dimensions are in $\mathcal{O}\left((r^p + dpr)\log(p)\right)$ and the generalization gap is in $\mathcal{O}\left(\sqrt{\frac{(r^p+dpr)\log(n)}{n}}\right)$ with high probability for any classifer parameterized by a low-rank Tucker tensor. It is worth observing that in contrast with the tensor train decomposition, all bounds have an exponential dependency on the tensor order $p$. In [41], the authors give an upper bound on the analogue of the growth function for tensor completion problems which is equivalent to ours. In [50], the pseudo-dimension of regression functions whose weight parameters have low Tucker rank is upper-bounded by $\mathcal{O}\left((r^p + drp)\log(pd^{p-1})\right)$, which is looser than our bound due to the term $d^{p-1}$ (though a similar argument to the one we use in the proof of Theorem 4 can be used to tighten the bound given in [50]).

**Tree tensor networks**   Lastly, we compare our result with the ones presented in [38] where the authors study the complexity of learning with tree tensor networks using metric entropy and covering numbers. The results presented in [38] only hold for TN structures whose underlying graph $G$ is a tree. Let $G$ be a tree and $\ell$ be a loss function which is both bounded and Lipschitz. Under these assumptions, it is shown in [38] that, for any $h \in \mathcal{H}^{regression}_G$, with high probability over the choice of a sample $S$ of size $n$ drawn i.i.d. from a distribution $D$, the generalization gap $R(h) - \hat{R}(h)$ is in $\tilde{\mathcal{O}}(\sqrt{N_G}/n)$. Theorem 4 gives a similar upper bound in $\tilde{\mathcal{O}}(\sqrt{N_G}/n)$ on the generalization gap of low-rank tensor classifiers. However, our results hold for *any* TN structure $G$. Thus, in contrast with our general upper bound (Theorem 2), the bounds from [38] cannot be applied to TN structures containing cycles such as tensor ring and PEPS.

## 5   Lower Bounds

We now present lower bounds on the VC and pseudo-dimensions of standard TN models: rank-one, CP, Tucker, TT and TR.

Table 1: Summary of our results for common TN structures. Both lower and upper bounds hold for the VC/pseudo-dimension of $\mathcal{H}_G^{\text{classif}}$, $\mathcal{H}_G^{\text{completion}}$ and $\mathcal{H}_G^{\text{regression}}$ for the corresponding TN structure $G$ (see Equations (4-6)). The upper bounds follow from applying our general upper bound (Theorem 2) to each TN structure. The lower bounds are proved for each TN structure specifically. Each lower bound is followed by the condition under which it holds in parenthesis (small font). Note that the two bounds for TT and TR hold for both TN structures.

| | rank one | CP | Tucker | TT / TR |
|---|---|---|---|---|
| Decomposition | (diagram) | (diagram) | (diagram) | (diagram) |
| Lower Bound (condition) | $(d-1)p$ | $rd$ $\;(r \le d^{p-1})$ | $r^p$ $\;(r \le d)$ | $r^2 d$ $\;(r \le d^{\lfloor \frac{p-1}{2}\rfloor}, p \ge 3)$ $\quad\frac{p(r^2 d-1)}{3}$ $\;(r=d, p/3 \in \mathbb{N})$ |
| Upper bound | $2dp\log(12p)$ | $2prd\log(12p)$ | $2(r^p + prd)\log(24p)$ | $2pr^2 d\log(12p)$ |

**Theorem 7.** *The VC-dimension and pseudo-dimension of the classification, regression and completion hypothesis classes defined in Equations (4-6) for the rank-one, CP, Tucker, TT and TR tensor network structures satisfy the lower bounds presented in Table 1.*

*These lower bounds show that the general upper bound of Theorem 2 is tight up to a $\mathcal{O}\left(\log(p)\right)$ factor for rank-one, TT and TR tensors and is tight up to a constant for low-rank matrices.*

The proof of this theorem can be found in Appendix A.2. These lower bounds show that our general upper bound is nearly optimal (up to a $\log$ factor in $p$) for rank-one, TT and TR tensors. Indeed, for rank-one tensors we have $(d-1)p \le \mathcal{C}^{\text{rank-one}} \le 2dp\log(12p)$ and for TT and TR tensors of rank $r = d$ whose order $p$ is a multiple of 3 we have $p(r^2 d - 1)/3 \le \mathcal{C}_r^{\text{TT/TR}} \le pr^2 d \cdot 2\log(12p)$, where $\mathcal{C}^{\text{rank-one}}$ (resp. $\mathcal{C}_r^{\text{TT/TR}}$) denotes any of the VC/pseudo-dimension of the regression, classification and completion hypothesis classe associated with rank-one tensors (resp. rank $r$ TT and TR tensors). In addition, the lower bound for the CP case shows that our general upper bounds are tight up to a constant for matrices. Indeed, for $p = 2$ and $r \le d$ the bounds for the CP case give $rd \le \mathcal{C}_r^{\text{matrix}} \le 20rd$ where $\mathcal{C}_r^{\text{matrix}}$ denotes the VC/pseudo-dimension of the hypothesis classes associated with $d \times d$ matrices of rank at most $r$.

## 6 Conclusion

We derived a general upper bound on the VC and pseudo-dimension of a large class of tensor models parameterized by *arbitrary* tensor network structures for classification, regression and completion. We showed that this general bound can be applied to obtain bounds on the complexity of relevant machine learning models such as matrix and tensor completion, trace regression and TT-based linear classifiers. In particular, our result leads to an improved upper bound on the VC-dimension of low-rank matrices for completion tasks. As a corollary of our results, we answer the open question listed in [13] on the VC-dimension of the MPS classification model introduced in [56]. To demonstrate the tightness of our general upper bound, we derived a series of lower bounds for specific TN structures, notably showing that our bound is tight up to a constant for low-rank matrix models for completion, regression and classification.

Future directions include deriving tighter upper bounds and/or lower bounds for the specific TN structures. This includes investigating whether our general upper bound can be tightened by removing the log factor in the number of vertices of the TN structure, deriving a stronger lower bound for CP (we conjecture our lower bound can be improved by a factor $p$ for CP), and loosening the condition under which our stronger lower bound holds for TT and TR (for TR, we conjecture that a lower bound of $\tilde{\Omega}(pr^2 d)$ holds for any $p \ge 3$ and $r \le d^k$ for some value of $k > 1$). Studying other complexity measures (e.g. Rademacher complexity) and extending recent data-dependant generalization bounds for overparameterized deep neural networks, such as the ones used in [4, 34], to TN learning models is worth pursuing. Finally, building upon the connection between the depth of convolutional arithmetic circuits and tensor network structures introduced in [14], it is interesting to connect our result on the VC-dimension of tensor networks to the expressiveness and generalization ability of neural networks.

**Acknowledgements**

We thank Maude Lizaire for feedbacks and the anonymous reviewers for the useful comments. This research is supported by the Canadian Institute for Advanced Research (CIFAR AI chair program) and the Natural Sciences and Engineering Research Council of Canada (Discovery program, RGPIN-2019-05949).

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
