# Lower and Upper Bounds on the Pseudo-Dimension of Tensor Network Models

# (Supplementary Material)

## A  Proofs

### A.1  Upper Bound and Generalization Bound

#### A.1.1  Proof of Theorem 2

**Theorem.** *Let $G = (V, E, \dim)$ be a tensor network structure and let $\mathcal{H}_G^{regression}$, $\mathcal{H}_G^{classif}$, $\mathcal{H}_G^{completion}$ be the corresponding hypothesis classes defined in Equations* (4)*,* (5)*,* (6) *where each model has $N_G$ parameters (see Equation* (3)*).*

*Then,* $\mathrm{Pdim}(\mathcal{H}_G^{regression})$, $d_{\mathrm{VC}}(\mathcal{H}_G^{classif})$ *and* $\mathrm{Pdim}(\mathcal{H}_G^{completion})$ *are all upper bounded by* $2N_G \log(12|V|)$.

*Proof.* We start with the pseudo-dimension introduced in Definition 1. Consider $n$ input tensors $\boldsymbol{\mathcal{X}}_1, \cdots, \boldsymbol{\mathcal{X}}_n$ and arbitrary threshold values $t_1, \cdots, t_n$. To upper-bound $\mathrm{Pdim}(\mathcal{H}_G^{regression})$, it is enough to show that for any set $S = \{\boldsymbol{\mathcal{X}}_1, \cdots, \boldsymbol{\mathcal{X}}_n\}$ and threshold values $t_1 \cdots, t_n$, the number of *relative sign patterns* realized by the class of functions $\mathcal{H}_G^{regression}$, is bounded by a value depending only on $n$ and tensor network structure $G$. Formally, we define the maximal number of sign patterns as follows:

$$f(n, G) := \sup_{\substack{\boldsymbol{\mathcal{X}}_1, \cdots \boldsymbol{\mathcal{X}}_n \in \mathcal{X} \\ t_1, \cdots, t_n \in \mathbb{R}}} \left| \left\{ \begin{pmatrix} \mathrm{sign}(h(\boldsymbol{\mathcal{X}}_1) - t_1) \\ \vdots \\ \mathrm{sign}(h(\boldsymbol{\mathcal{X}}_n) - t_n) \end{pmatrix} \mid h \in \mathcal{H}_G^{regression} \right\} \right| \tag{8}$$

For $h \in \mathcal{H}_G^{regression}$, by definition, $h : \boldsymbol{\mathcal{X}} \mapsto \langle \boldsymbol{\mathcal{W}}, \boldsymbol{\mathcal{X}} \rangle$ for some weight tensor $\boldsymbol{\mathcal{W}} \in \mathcal{T}(G)$. Consequently, there exists a collection of core tensors $\boldsymbol{\mathcal{T}}^v \in \bigotimes_{e \in E_v} \mathbb{R}^{dim(e)}$ such that $\boldsymbol{\mathcal{W}} = TN(G, \{\boldsymbol{\mathcal{T}}^v\}_{v \in V})$ (see Equation (2)) and it follows that $h(\boldsymbol{\mathcal{X}})$ is a polynomial of degree $|V|$ over $N_G$ variables. The variables of the polynomial are the entries of the core tensors $\{\boldsymbol{\mathcal{T}}^v\}_{v \in V}$.

Now, given a set of input tensors $S = \{\boldsymbol{\mathcal{X}}_1, \cdots, \boldsymbol{\mathcal{X}}_n\}$, the value $f(n, G)$ in Equation (8) is thus bounded by the number of sign patterns that a system of $n$ polynomial equations (one for each input data point) of order $|V|$ over $N_G$ variables can take. It then follows from Warren's theorem (Theorem 3) that

$$f(n, G) \leq \left( \frac{4en|V|}{N_G} \right)^{N_G}. \tag{9}$$

**Bound on the pseudo-dimension**  To extract a bound on the pseudo-dimension from the above bound on the number of *relative sign patterns*, we follow the line of the proof of Theorem 8.3 in [3]. First observe that by the definition of the pseudo-dimension, if $f(n, G) < 2^n$ for some $n$, then $\mathrm{Pdim}(\mathcal{H}_G^{regression}) < n$. Using the bound on $f(n, G)$, we have $f(n, G) \leq \left( \frac{4en|V|}{N_G} \right)^{N_G} < 2^n$ if and only if

$$N_G \left( \log n + \log \frac{4e|V|}{N_G} \right) < n, \tag{10}$$

with the logarithm being in base 2. Using the classical inequality $\ln n \leq nb + \ln \frac{1}{b} - 1$, or equivalently $\log n \leq \frac{nb}{\ln 2} + \log \frac{1}{eb}$, it follows that

$$\log n \leq \frac{n}{2N_G} + \log \frac{2N_G}{e \ln 2}.$$

Consequently, Equation (10) is implied by $n > 2N_G \log \frac{8|V|}{\ln 2}$, which is in turn implied by $n > 2N_G \log(12|V|)$.

We thus have shown that $\text{Pdim}(\mathcal{H}_G^{\text{regression}}) \leq 2N_G \log(12|V|)$. Since $\text{Pdim}(\mathcal{H}) = d_{\text{VC}}(\{(x,t) \mapsto \text{sign}(h(x)-t) \mid h \in \mathcal{H}\})$ for any hypothesis class $\mathcal{H}$, this upper bound implies that there exists no set of $k \geq 2N_G \log(12|V|)$ points that are shattered by the hypothesis class

$$\{(\boldsymbol{\mathcal{X}},t) \mapsto \text{sign}(h(\boldsymbol{\mathcal{X}})-t) \mid h \in \mathcal{H}_G^{\text{regression}}\} = \{(\boldsymbol{\mathcal{X}},t) \mapsto \text{sign}(\langle \boldsymbol{\mathcal{W}}, \boldsymbol{\mathcal{X}}\rangle - t) \mid \boldsymbol{\mathcal{W}} \in \boldsymbol{\mathcal{T}}(G)\}.$$

In particular, no set of $k$ points with thresholds $t_1 = \cdots = t_k = 0$ is shattered by $\mathcal{H}_G^{\text{regression}}$, which is equivalent to no set of $k$ points being shattered by $\mathcal{H}_G^{\text{classif}}$, hence $d_{\text{VC}}(\mathcal{H}_G^{\text{classif}}) \leq 2N_G \log(12|V|)$.

Similarly, for the completion case we argue that the maximum number of multiples of indices shattered by the function class $\mathcal{H}_G^{\text{completion}}$ is bounded by the same value as $\text{Pdim}(\mathcal{H}_G^{\text{regression}})$. The *Pseudo-dimension* of $\mathcal{H}_G^{\text{completion}}$ is by definition, the maximum number of indices, i.e., the maximum number of the entries of the tensor, that could be pseudo-shattered (with thresholds zero) by the class of tensors $\mathcal{H}_G^{\text{completion}}$. Each component of the tensor $\boldsymbol{\mathcal{W}} \in \boldsymbol{\mathcal{T}}(G)$, i.e., $\boldsymbol{\mathcal{W}}_{i_1,\cdots i_p}$, can be written as the following inner product

$$\boldsymbol{\mathcal{W}}_{i_1,\cdots i_p} = \langle \boldsymbol{\mathcal{W}}, \mathbf{e}_{i_1}^{(1)} \otimes \mathbf{e}_{i_2}^{(2)} \otimes \cdots \otimes \mathbf{e}_{i_p}^{(p)}\rangle$$

where each $\mathbf{e}_i^{(j)} \in \mathbb{R}^{d_j}$ is the $i$-th basis vector of the standard basis of $\mathbb{R}^{d_j}$. By this way of writing, we make a clear analogy with a regression problem where data points are , $\mathbf{e}_{i_1}^{(1)} \otimes \mathbf{e}_{i_2}^{(2)} \otimes \cdots \otimes \mathbf{e}_{i_p}^{(p)}$ and the weight tensor is $\boldsymbol{\mathcal{W}}$.

According to this analogy, no set of more than $2N_G \log(12|V|)$ indices can be shattered by $\mathcal{H}_G^{\text{completion}}$; since otherwise we would have a set of data points $\mathbf{e}_{i_1}^{(1)} \otimes \cdots \otimes \mathbf{e}_{i_p}^{(p)}$ that is shattered by $\mathcal{H}_G^{\text{regression}}$, which contradicts our earlier result $\text{Pdim}(\mathcal{H}_G^{\text{regression}}) \leq 2N_G \log(12|V|)$. Therefore, we conclude that $\text{Pdim}(\mathcal{H}_G^{\text{completion}}) \leq 2N_G \log(12|V|)$. □

### A.1.2 Proof of Theorem 4

**Theorem.** *Let $S$ be a sample of size $n$ drawn from a distribution $D$. Then, for any $\delta > 0$, with probability at least $1 - \delta$ over the choice of $S$, we have for any $h \in \mathcal{H}_G^{classif}$*

$$R(h) < \hat{R}_S(h) + 2\sqrt{\frac{2}{n}\left(N_G \log \frac{8en|V|}{N_G} + \log \frac{4}{\delta}\right)}. \tag{11}$$

*Proof.* We use a symmetrization lemma and a corollary of Hoeffding's inequality. For this part, let $\mathcal{H} = \mathcal{H}_G^{\text{classif}}$.

**Lemma 8.** *(Symmetrization Lemma) [6, Lemma 2] Let $S$ and $S'$ be two random samples of size $n$ drawn from a distribution $D$. Then for any $t > 0$ such that $nt^2 \geq 2$, we have*

$$\mathbb{P}_{S\sim D}\left[\sup_{h\in\mathcal{H}}\left(R(h) - \hat{R}_S(h)\right) \geq t\right] \leq 2\,\mathbb{P}_{S,S'\sim D}\left[\sup_{h\in\mathcal{H}}\left(\hat{R}_{S'}(h) - \hat{R}_S(h)\right) \geq \frac{t}{2}\right], \tag{12}$$

**Corollary 9.** *If $Z_1,\ldots,Z_n,Z'_1,\ldots,Z'_n$ are $2n$ i.i.d. random variables with values in $[0,1]$, then for all $\varepsilon > 0$ we have*

$$\mathbb{P}\left[\frac{1}{n}\sum_{i=1}^n Z_i - \frac{1}{n}\sum_{i=1}^n Z'_i > \varepsilon\right] \leq 2\exp\left(-\frac{n\varepsilon^2}{2}\right) \tag{13}$$

This statement is proved by rewriting $\mathbb{P}[\frac{1}{n}\sum_{i=1}^n Z_i - \frac{1}{n}\sum_{i=1}^n Z'_i > \varepsilon]$ as $\mathbb{P}[\frac{1}{n}\sum_{i=1}^n(Z_i - \mathbb{E}[Z_i]) - \frac{1}{n}\sum_{i=1}^n(Z'_i - \mathbb{E}[Z'_i]) > \varepsilon] \leq \mathbb{P}[\frac{1}{n}\sum_{i=1}^n(Z_i - \mathbb{E}[Z_i]) > \frac{\varepsilon}{2}] + \mathbb{P}[\frac{1}{n}\sum_{i=1}^n(-Z'_i + \mathbb{E}[Z'_i]) > \frac{\varepsilon}{2}]$. Then by using Hoeffding's inequality, Equation (13) is proved.

From Lemma 8 we have

$$\mathbb{P}_{S\sim D}\left[\sup_{h\in\mathcal{H}}\left(R(h) - \hat{R}_S(h)\right) \geq 2\varepsilon\right] \leq 2\mathbb{P}_{S\sim D,S'\sim D}\left[\max_{h\in\mathcal{H}_{S,S'}}\left(\hat{R}_{S'}(h) - \hat{R}_S(h)\right) \geq \varepsilon\right]$$

$$\leq 2\,\mathbb{P}_{S,S'\sim D}\left[\exists h \in \mathcal{H}_{S,S'} \mid \left(\hat{R}_{S'}(h) - \hat{R}_S(h)\right) \geq \varepsilon\right], \tag{14}$$

where $\mathcal{H}_{S,S'}$ is the projection of the hypothesis class $\mathcal{H}$ onto the subset $S \cup S'$. Then, by applying the union bound followed by Corollary 9 (by taking the bounded loss as the random variable $Z$) and recalling the notion of the growth function from Definition 1, we get

$$\mathbb{P}_{S \sim D}[\sup_{h \in \mathcal{H}} \left( R(h) - \hat{R}_S(h) \right) \geq 2\varepsilon] \leq 2\Pi_{\mathcal{H}}(2n) \left( \mathbb{P}_{S,S' \sim D} \left[ \hat{R}_{S'}(h) - \hat{R}_S(h) \geq \varepsilon \right] \right)$$

$$\leq 4 \, \Pi_{\mathcal{H}}(2n) \exp\left(-\frac{n\varepsilon^2}{2}\right), \tag{15}$$

In order to upper bound the growth function of the hypothesis class $\mathcal{H}_G^{\text{classif}}$ we can use the same argument as we did for the pseudo-dimension, which results in an upper bound similar to the one for the *number of relative sign patterns* in Equation (9)

$$\Pi_{\mathcal{H}_{\mathcal{G}}^{\text{classif}}}(n) \leq \left( \frac{4en|V|}{N_G} \right)^{N_G} \tag{16}$$

Combining Equations (15) and (16), we get

$$\mathbb{P}_{S \sim D} \left[ \sup_{h \in \mathcal{H}} \left( R(h) - \hat{R}_S(h) \right) \geq \varepsilon \right] \leq 4 \left( \frac{8en|V|}{N_G} \right)^{N_G} e^{-\frac{n\varepsilon^2}{8}} \tag{17}$$

Equation (11) then directly follows from setting the failure probability equal to $\delta$ and solving for $\varepsilon$. $\qquad\square$

## A.2 Proof of Theorem 7

In this section, we give the proofs of all the lower bounds appearing in Table 1. All the proofs rely on the following lemma which gives a useful way for jointly deriving lower bounds on the pseudo-dimension and VC-dimension of the hypothesis classes of linear models for regression, completion and classification defined in Eq. (4-6).

**Lemma 10.** *Let $V \subset \mathbb{R}^d$ and define the hypothesis classes*

$$\mathcal{H}^{completion} = \{h : i \mapsto \mathbf{w}_i \mid \mathbf{w} \in V\}$$

$$\mathcal{H}^{regression} = \{h : \mathbf{x} \mapsto \langle \mathbf{w}, \mathbf{x} \rangle \mid \mathbf{w} \in V\}$$

$$\mathcal{H}^{classif} = \{h : \mathbf{x} \mapsto \text{sign}(\langle \mathbf{w}, \mathbf{x} \rangle) \mid \mathbf{w} \in V\}.$$

*If there exist $k$ indices $i_1, \cdots, i_k \in [d]$ that are shattered by $V$, i.e., such that*

$$|\{(\text{sign}(\mathbf{w}_{i_1}), \text{sign}(\mathbf{w}_{i_2}), \cdots, \text{sign}(\mathbf{w}_{i_k})) \mid \mathbf{w} \in V\}| = 2^k,$$

*then $d_{\text{VC}}(\mathcal{H}^{classif})$, $\text{Pdim}(\mathcal{H}^{regression})$ and $\text{Pdim}(\mathcal{H}^{completion})$ are all lower-bounded by $k$.*

*Proof.* Let $\mathbf{e}_1, \cdots, \mathbf{e}_d$ be the canonical basis of $\mathbb{R}^d$ and let $i_1, \cdots, i_k \in [d]$ be a set of indices shattered by $V$. Since $\langle \mathbf{w}, \mathbf{e}_i \rangle = \mathbf{w}_i$ for all $i \in [d]$, the points $\mathbf{e}_{i_1}, \cdots, \mathbf{e}_{i_k}$ are shattered by $\mathcal{H}^{\text{classif}}$ and thus $d_{\text{VC}}(\mathcal{H}^{\text{classif}}) \geq k$.

Similarly, since $\text{Pdim}(\mathcal{H}) = d_{\text{VC}}(\{(x,t) \mapsto \text{sign}(h(x) - t) \mid h \in \mathcal{H}\})$ for any hypothesis class $\mathcal{H}$, the set of points $\mathbf{e}_{i_1}, \cdots, \mathbf{e}_{i_k}$ with thresholds $t_1 = t_2 = \cdots = t_k = 0$ is shattered by the hypothesis class $\{(\mathbf{x}, t) \mapsto \text{sign}(\langle \mathbf{w}, \mathbf{x} \rangle - t) \mid \mathbf{w} \in V\}$, and thus $\text{Pdim}(\mathcal{H}^{\text{regression}}) \geq k$.

Lastly, the set of indices $i_1, \cdots, i_k$ with thresholds $t_1 = t_2 = \cdots = t_k = 0$ is shattered by the class $\{(i, t) \mapsto \text{sign}(\mathbf{w}_i - t) \mid \mathbf{w} \in V\}$, and thus $\text{Pdim}(\mathcal{H}^{\text{completion}}) \geq k$. $\qquad\square$

### A.2.1 Rank-One Tensors

**Theorem 11.** *Let $G_{rank\text{-}one} = \overset{d}{\bullet}\,\overset{d}{\bullet}\, \cdots \,\overset{d}{\bullet}\,\overset{d}{\bullet}$ be the tensor network structure corresponding to $p$-th order rank-one tensors, i.e., $\mathcal{T}(G_{rank\text{-}one}) = \{\mathbf{u}_1 \otimes \mathbf{u}_2 \otimes \cdots \otimes \mathbf{u}_p \mid \mathbf{u}_1, \mathbf{u}_2, \cdots, \mathbf{u}_p \in \mathbb{R}^d\}$.*

*The VC-dimension and pseudo-dimensions $d_{\text{VC}}(\mathcal{H}_{G_{rank\text{-}one}}^{classif})$, $\text{Pdim}(\mathcal{H}_{G_{rank\text{-}one}}^{regression})$, $\text{Pdim}(\mathcal{H}_{G_{rank\text{-}one}}^{completion})$ are all lower-bounded by $(d-1)p$.*

*Proof.* We show that the set of indices

$$S = \{(\underbrace{d, \cdots, d}_{i-1 \text{ times}}, j, \underbrace{d, \cdots, d}_{p-i \text{ times}}) \mid i \in [p], j \in [d-1]\}$$

is shattered by $\mathcal{T}(G_{\text{rank-one}})$, the result then follows from Lemma 10. More precisely, we show that $S$ is shattered by the set of rank-one tensors

$$A = \left\{ \begin{pmatrix} \mathbf{v}_1 \\ 1 \end{pmatrix} \otimes \begin{pmatrix} \mathbf{v}_2 \\ 1 \end{pmatrix} \otimes \cdots \otimes \begin{pmatrix} \mathbf{v}_p \\ 1 \end{pmatrix} \mid \mathbf{v}_1, \mathbf{v}_2, \cdots, \mathbf{v}_p \in \mathbb{R}^{d-1} \right\} \subset \mathcal{T}(G_{\text{rank-one}}).$$

Indeed, for any multi-index $(\underbrace{d, \cdots, d}_{i-1 \text{ times}}, j, \underbrace{d, \cdots, d}_{p-i \text{ times}}) \in S$ and any rank-one tensor $\mathcal{X} = \begin{pmatrix} \mathbf{v}_1 \\ 1 \end{pmatrix} \otimes \begin{pmatrix} \mathbf{v}_2 \\ 1 \end{pmatrix} \otimes \cdots \otimes \begin{pmatrix} \mathbf{v}_p \\ 1 \end{pmatrix} \in A$, we have

$$\mathcal{X}_{\underbrace{d, \cdots, d}_{i-1 \text{ times}}, j, \underbrace{d, \cdots, d}_{p-i \text{ times}}} = \left( \begin{pmatrix} \mathbf{v}_1 \\ 1 \end{pmatrix} \otimes \begin{pmatrix} \mathbf{v}_2 \\ 1 \end{pmatrix} \otimes \cdots \otimes \begin{pmatrix} \mathbf{v}_p \\ 1 \end{pmatrix} \right)_{d, \cdots, d, j, d, \cdots, d} = (\mathbf{v}_i)_j.$$

It follows that the $(d-1)p$ components $\mathcal{X}_{i_1, \cdots, i_p}$ for $\mathcal{X} \in A$ and $(i_1, \cdots, i_p) \in S$ can take any arbitrary values (the entries of the vectors $\mathbf{v}_1, \cdots, \mathbf{v}_p \in \mathbb{R}^{d-1}$) and thus, that $S$ is shattered by $A$ and accordingly by $\mathcal{T}(G_{\text{rank-one}})$. The result then directly follows from Lemma 10. $\square$

### A.2.2 Tensor Train and Tensor Ring

**Theorem 12.** *Let* $r \leq d^{\lfloor \frac{p-1}{2} \rfloor}$, *let* $G_{TT}(r) = $  *be the tensor network structure corresponding to $p$-th order tensors of tensor train rank at most $r$, and let* $G_{TR}(r) = $  *be the tensor network structure corresponding to $p$-th order tensors of tensor ring rank at most $r$.*

*Then, the VC-dimension and pseudo-dimensions* $d_{\text{VC}}(\mathcal{H}_{G_{TT}(r)}^{classif})$, $d_{\text{VC}}(\mathcal{H}_{G_{TR}(r)}^{classif})$, $\text{Pdim}(\mathcal{H}_{G_{TT}(r)}^{regression})$, $\text{Pdim}(\mathcal{H}_{G_{TR}(r)}^{regression})$, $\text{Pdim}(\mathcal{H}_{G_{TT}(r)}^{completion})$ *and* $\text{Pdim}(\mathcal{H}_{G_{TR}(r)}^{completion})$ *are all lower-bounded by* $r^2 d$.

*Moreover, in the particular case where* $r = d$ *and* $p = 3k$ *for some* $k \in \mathbb{N}$, *the VC-dimension and pseudo-dimensions* $d_{\text{VC}}(\mathcal{H}_{G_{TT}(r)}^{classif})$, $d_{\text{VC}}(\mathcal{H}_{G_{TR}(r)}^{classif})$, $\text{Pdim}(\mathcal{H}_{G_{TT}(r)}^{regression})$, $\text{Pdim}(\mathcal{H}_{G_{TR}(r)}^{regression})$, $\text{Pdim}(\mathcal{H}_{G_{TT}(r)}^{completion})$ *and* $\text{Pdim}(\mathcal{H}_{G_{TR}(r)}^{completion})$ *are all lower-bounded by* $\frac{p(r^2 d - 1)}{3}$.

*Proof.* We start with the tensor train case, the tensor ring case will be handled similarly.

Let $r \leq d^{\lfloor \frac{p-1}{2} \rfloor}$. We will show that there exists a set of $r^2 d$ indices $(i_1, \cdots, j_1), \cdots, (i_{r^2 d}, \cdots, j_{r^2 d})$ that is shattered by $\mathcal{T}(G_{TT}(r))$ (the set of tensors of tensor train rank at most $r$), i.e., such that

$$\left| \{(\text{sign}(\mathcal{W}_{i_1, \cdots, j_1}), \text{sign}(\mathcal{W}_{i_2, \cdots, j_2}), \cdots, \text{sign}(\mathcal{W}_{i_{r^2 d}, \cdots, j_{r^2 d}})) \mid \mathcal{W} \in \mathcal{T}(G_{TT}(r))\} \right| = 2^{r^2 d}.$$

In order to do so, we will consider a tensor train tensor $\mathcal{T}$ with cores $\mathcal{G}^{(1)}, \cdots, \mathcal{G}^{(p)}$, where one of the core tensors will be free, while the other cores are fixed in such a way that each component of the free core tensor appears exactly once in the entries of $\mathcal{T}$.

Let $\mathbf{e}_1, \cdots, \mathbf{e}_r$ be the canonical basis of $\mathbb{R}^r$ and let $\mathbf{e}_i = \mathbf{0}$ for any $i > r$. Let $k = \lfloor \frac{p}{2} \rfloor$ and let $\mathcal{G}^{(k)}$ be the $k$-th core of the tensor train tensor $\mathcal{T}$ (i.e., the middle core). The other cores of $\mathcal{T}$ are defined as follows: for each $j \in [d]$,

$$\mathcal{G}^{(1)}_{j,:} = \mathbf{e}_j^\top$$

$$\mathcal{G}^{(s)}_{:,j,:} = \mathbf{e}_1 \mathbf{e}_{(j-1)d^{s-1}+1}^\top + \mathbf{e}_2 \mathbf{e}_{(j-1)d^{s-1}+2}^\top + \cdots + \mathbf{e}_r \mathbf{e}_{(j-1)d^{s-1}+r}^\top \qquad \text{for } s = 2, \cdots, k-1$$

$$\mathcal{G}^{(s)}_{:,j,:} = \mathbf{e}_{(j-1)d^{p-s}+1} \mathbf{e}_1^\top + \mathbf{e}_{(j-1)d^{p-s}+2} \mathbf{e}_2^\top + \cdots + \mathbf{e}_{(j-1)d^{p-s}+r} \mathbf{e}_r^\top \qquad \text{for } s = k+1, \cdots, p-1$$

$$\mathcal{G}^{(p)}_{:,j} = \mathbf{e}_j.$$

With these definitions, one can check that

$$\mathcal{G}_{i_1,:}^{(1)}\mathcal{G}_{:,i_2,:}^{(2)}\mathcal{G}_{:,i_3,:}^{(3)}\cdots\mathcal{G}_{:,i_{k-1},:}^{(k-1)} = \mathbf{e}_{i_1+(i_2-1)d+(i_3-1)d^2+\cdots+(i_{k-1}-1)d^{k-2}}^{\top}$$

for any $i_1,\cdots,i_{k-1} \in [d]$ and

$$\mathcal{G}_{:,i_{k+1},:}^{(k+1)}\mathcal{G}_{:,i_{k+2},:}^{(k+2)}\cdots\mathcal{G}_{:,i_{p-1},:}^{(p-1)}\mathcal{G}_{:,i_p}^{(p)} = \mathbf{e}_{i_p+(i_{p-1}-1)d+(i_{p-2}-1)d^2+\cdots+(i_{k+1}-1)d^{p-k-1}}$$

for any $i_{k+1},\cdots,i_p \in [d]$. Letting $[\![j_0,\cdots,j_t]\!] = j_0 + (j_1-1)d + (j_2-1)d^2 + \cdots + (j_t-1)d^t$ for any $j_0,\cdots,j_t \in [d]$, it follows that for any $i_1,\cdots,i_p \in [d]$,

$$\mathcal{T}_{i_1,\cdots,i_p} = \begin{cases} \mathcal{G}_{[\![i_1,i_2\cdots,i_{k-1}]\!],i_k,[\![i_p,i_{p-1},\cdots,i_{k+1}]\!]}^{(k)} & \text{if } [\![i_1,i_2\cdots,i_{k-1}]\!] \le r \text{ and } [\![i_p,i_{p-1},\cdots,i_{k+1}]\!] \le r \\ 0 & \text{otherwise.} \end{cases}$$

Since $r \le d^{\lfloor\frac{p-1}{2}\rfloor}$ and $k = \lfloor\frac{p}{2}\rfloor$, this implies that for any $k$-th core $\mathcal{G}^{(k)}$, the tensor train tensor $\mathcal{T}$ contains all the $r^2 d$ entries of $\mathcal{G}^{(k)}$. Thus, the set of $r^2 d$ indices $\{(i_1,\cdots,i_p) \mid [\![i_1,i_2\cdots,i_{k-1}]\!] \le r, \ i_k \in [d], [\![i_p,i_{p-1},\cdots,i_{k+1}]\!] \le r\}$ is shattered by $\mathcal{T}(G_{\mathrm{TT}}(r))$ and the first part of the theorem follows from Lemma 10.

We now prove the second part of the theorem for the TT case, using a different construction. Let $r = d$ and $p = 3k$ for some $k \in \mathbb{N}$. We will construct a family of tensors in $\mathcal{T}(G_{\mathrm{TT}}(r))$ where a third of the $p = 3k$ cores will be free while the other cores are fixed in such a way that the resulting tensor $\mathcal{T}$ can be seen as the outer product of $k$ 3rd order tensor of size $d \times d \times d$. By observing that such tensors can be interpreted as rank-one $k$-th order tensors in $\mathbb{R}^{d^3 \times d^3 \times \cdots \times d^3}$, the second part of the theorem will follow from Theorem 11.

Let $\mathcal{G}^{(1)},\cdots,\mathcal{G}^{(p)}$ be the core tensors of the TT decomposition. The core tensors $\mathcal{G}^{(3s+2)} \in \mathbb{R}^{d \times d \times d}$ for $s = 0,\cdots,k-1$ are free while the other cores are defined as follows: for any $j \in [d]$,

$$\mathcal{G}_{j,:}^{(1)} = \mathbf{e}_j^{\top}$$
$$\mathcal{G}_{:,j,:}^{(3s+3)} = \mathbf{e}_j\mathbf{e}_1^{\top} \qquad\qquad \text{for } s = 0,\cdots,k-2$$
$$\mathcal{G}_{:,j,:}^{(3s+1)} = \mathbf{e}_1\mathbf{e}_j^{\top} \qquad\qquad \text{for } s = 1,\cdots,k-1$$
$$\mathcal{G}_{j,:}^{(p)} = \mathbf{e}_j.$$

It follows that, for any $i_1,\cdots,i_p \in [d]$, we have

$$\begin{aligned}
\mathcal{T}_{i_1,\cdots,i_p} &= \mathcal{G}_{i_1,:}^{(1)}\mathcal{G}_{:,i_2,:}^{(2)}\cdots\mathcal{G}_{:,i_{p-1},:}^{(p-1)}\mathcal{G}_{:,i_p}^{(p)} \\
&= (\mathbf{e}_{i_1}^{\top})(\mathcal{G}_{:,i_2,:}^{(2)})(\mathbf{e}_{i_3}\mathbf{e}_1^{\top})\,(\mathbf{e}_1\mathbf{e}_{i_4}^{\top})(\mathcal{G}_{:,i_5,:}^{(5)})(\mathbf{e}_{i_6}\mathbf{e}_1^{\top})\cdots(\mathbf{e}_1\mathbf{e}_{i_{p-2}}^{\top})(\mathcal{G}_{:,i_{p-1},:}^{(p-1)})(\mathbf{e}_{i_p}) \\
&= \mathcal{G}_{i_1,i_2,i_3}^{(2)}\mathcal{G}_{i_4,i_5,i_6}^{(5)}\cdots\mathcal{G}_{i_{p-2},i_{p-1},i_p}^{(p-1)}
\end{aligned}$$

which implies that $\mathcal{T} = \mathcal{G}^{(2)} \otimes \mathcal{G}^{(5)} \otimes \cdots \otimes \mathcal{G}^{(p-1)} = \bigotimes_{s=0}^{k-1}\mathcal{G}^{(3s+2)}$. By reshaping the set of tensors constructed in this way into $k$th order tensors in $\mathbb{R}^{d^3 \times \cdots \times d^3}$, one can see that this set of tensors is exactly the set of rank one $k$th order tensors of size $d^3 \times \cdots \times d^3$, for which the corresponding VC dimension and pseudo dimensions are lower bounded by $k(d^3-1) = p(r^2d-1)/3$ from Theorem 11.

The proof for the tensor ring case uses the exact same constructions with the difference in the definition of the first and last core tensors which are defined by $\mathcal{G}_{:,j,:}^{(1)} = \mathbf{e}_1\mathbf{e}_j^{\top}$ and $\mathcal{G}_{:,j,:}^{(p)} = \mathbf{e}_j\mathbf{e}_1^{\top}$ for each $j \in [d]$. With these definitions, one can check that

$$\mathcal{G}_{:,i_1,:}^{(1)}\mathcal{G}_{:,i_2,:}^{(2)}\mathcal{G}_{:,i_3,:}^{(3)}\cdots\mathcal{G}_{:,i_{k-1},:}^{(k-1)} = \mathbf{e}_1\mathbf{e}_{[\![i_1,i_2,\cdots i_{k-1}]\!]}^{\top}$$

for any $i_1,\cdots,i_{k-1} \in [d]$ and

$$\mathcal{G}_{:,i_{k+1},:}^{(k+1)}\mathcal{G}_{:,i_{k+2},:}^{(k+2)}\cdots\mathcal{G}_{:,i_{p-1},:}^{(p-1)}\mathcal{G}_{:,i_p,:}^{(p)} = \mathbf{e}_{[\![i_p,i_{p-1},i_{k+1}]\!]}\mathbf{e}_1^{\top}$$

for any $i_{k+1}, \cdots, i_p \in [d]$. It follows that for any $i_1, \cdots, i_p \in [d]$,

$$\mathcal{T}_{i_1, \cdots, i_p} = \mathrm{Tr}\left(\mathcal{G}^{(1)}_{:,i_1,:} \mathcal{G}^{(2)}_{:,i_2,:} \mathcal{G}^{(3)}_{:,i_3,:} \cdots \mathcal{G}^{(k-1)}_{:,i_{k-1},:} \mathcal{G}^{(k)} \mathcal{G}^{(k+1)}_{:,i_{k+1},:} \mathcal{G}^{(k+2)}_{:,i_{k+2},:} \cdots \mathcal{G}^{(p-1)}_{:,i_{p-1},:} \mathcal{G}^{(p)}_{:,i_p,:}\right)$$

$$= \begin{cases} \mathcal{G}^{(k)}_{[\![i_1,i_2\cdots,i_{k-1}]\!],i_k,[\![i_p,i_{p-1},\cdots,i_{k+1}]\!]} & \text{if } [\![i_1,i_2\cdots,i_{k-1}]\!] \leq r \text{ and } [\![i_p,i_{p-1},\cdots,i_{k+1}]\!] \leq r \\ 0 & \text{otherwise.} \end{cases}$$

The proof of the first part of the theorem then follows the exact same argument as for the TT case. The second part of the theorem for TR is proved exactly as the one for TT by replacing the first and last cores again by $\mathcal{G}^{(1)}_{:,j,:} = \mathbf{e}_1 \mathbf{e}_j^\top$ and $\mathcal{G}^{(p)}_{:,j,:} = \mathbf{e}_j \mathbf{e}_1^\top$ for each $j \in [d]$.

$\square$

### A.2.3  Tucker

**Theorem 13.** *Let $r \leq d$ and let $G_{Tucker}(r) = $*  *be the tensor network structure corresponding to $p$-th order tensors of Tucker rank at most $r$.*

*Then, the VC-dimension and pseudo-dimensions $d_{\mathrm{VC}}(\mathcal{H}^{classif}_{G_{Tucker}(r)})$, $\mathrm{Pdim}(\mathcal{H}^{regression}_{G_{Tucker}(r)})$ and $\mathrm{Pdim}(\mathcal{H}^{completion}_{G_{Tucker}(r)})$ are all lower bounded by $r^p$.*

*Proof.* Let $r \leq d$. We show that there exists a set of $r^p$ indices $(i_1, \cdots, j_1), \cdots, (i_{r^p}, \cdots, j_{r^p})$ that is shattered by $\mathcal{T}(G_{Tucker}(r))$ (the set of tensors of Tucker rank at most $r$), i.e., such that

$$|\{(\mathrm{sign}(\mathcal{W}_{i_1,\cdots,j_1}), \mathrm{sign}(\mathcal{W}_{i_2,\cdots,j_2}), \cdots, \mathrm{sign}(\mathcal{W}_{i_{r^p},\cdots,j_{r^p}})) \mid \mathcal{W} \in \mathcal{T}(G_{Tucker}(r))\}| = 2^{r^p}.$$

Let $\mathbf{P} = \begin{pmatrix} \mathbf{I}_{r \times r} & \mathbf{0}_{r \times (d-r)} \end{pmatrix}^\top \in \mathbb{R}^{d \times r}$. We consider the following subset of $\mathcal{T}(G_{Tucker}(r))$:

$$A = \{\mathcal{G} \times_1 \mathbf{P} \times_2 \mathbf{P} \times_3 \cdots \times_p \mathbf{P} \mid \mathcal{G} \in \mathbb{R}^{r \times r \times \cdots \times r}\} \subset \mathcal{T}(G_{Tucker}(r))$$

where $\times_k$ denotes the mode-$k$ product (see, e.g., [32]). It is easy to see that any tensor $\mathcal{T} = \mathcal{G} \times_1 \mathbf{P} \times_2 \mathbf{P} \times_3 \cdots \times_p \mathbf{P} \in A$ will have entries $\mathcal{T}_{i_1,\cdots,i_p} = \mathcal{G}_{i_1,\cdots,i_p}$ for any $i_1, \cdots, i_p \in [r]$. Hence the set of $r^p$ indices $[r] \times [r] \times \cdots \times [r] \subset [d] \times [d] \times \cdots \times [d]$ is shattered by $\mathcal{T}(G_{Tucker}(r))$ and the result directly follows from Lemma 10. $\square$

### A.2.4  CP

**Theorem 14.** *Let $r \leq d^{p-1}$ and let $G_{CP}(r) = $*  *be the tensor network structure corresponding to $p$-th order tensors of CP rank at most $r$.*

*Then, the VC-dimension and pseudo-dimensions $d_{\mathrm{VC}}(\mathcal{H}^{classif}_{G_{CP}(r)})$, $\mathrm{Pdim}(\mathcal{H}^{regression}_{G_{CP}(r)})$ and $\mathrm{Pdim}(\mathcal{H}^{completion}_{G_{CP}(r)})$ are all lower-bounded by $rd$.*

*Proof.* Let $r \leq d^{p-1}$. We show that there exists a set of $rd$ indices $(i_1, \cdots, j_1), \cdots, (i_{rd}, \cdots, j_{rd})$ that is shattered by $\mathcal{T}(G_{CP}(r))$ (the set of tensors of CP rank at most $r$), i.e., such that

$$|\{(\mathrm{sign}(\mathcal{W}_{i_1,\cdots,j_1}), \mathrm{sign}(\mathcal{W}_{i_2,\cdots,j_2}), \cdots, \mathrm{sign}(\mathcal{W}_{i_{rd},\cdots,j_{rd}})) \mid \mathcal{W} \in \mathcal{T}(G_{CP}(r))\}| = 2^{rd}.$$

We construct a tensor $\mathcal{T}$ of CP rank at most $r$ such that each component of a matrix $\mathbf{A} \in \mathbb{R}^{d \times r}$ appears at least once in the entries of $\mathcal{T}$. Similarly to the previous proofs, $\mathbf{A}$ will be a free parameter allowed to take any value while the other components of the parametrization of $\mathcal{T}$ will be fixed.

Let $\mathbf{A} \in \mathbb{R}^{d \times r}$, we define $p$ tensors $\mathcal{A}^{(1)}, \cdots, \mathcal{A}^{(p)} \in \mathbb{R}^{d \times \cdots \times d}$ of order $p$ as follows: for all $i_1, \cdots, i_p, \tau_1, \cdots, \tau_{p-1} \in [d]$,

$$\mathcal{A}^{(1)}_{i_1,\tau_1,\cdots\tau_{p-1}} = \begin{cases} \mathbf{A}_{i_1, \tau_1 + (\tau_2-1)d + \cdots + (\tau_{p-1}-1)d^{p-2}} & \text{if } \tau_1 + (\tau_2 - 1)d + \cdots + (\tau_{p-1}-1)d^{p-2} \leq r \\ 0 & \text{otherwise} \end{cases}$$

$$\mathcal{A}^{(s)}_{i_s,\tau_1,\cdots\tau_{p-1}} = \delta_{i_s,\tau_{s-1}} \quad \text{for } s = 2, \cdots, p$$

where $\delta$ is the Kronecker symbol. Let $S = \{(\tau_1, \cdots, \tau_{p-1}) \in [d] \times \cdots \times [d] \mid \tau_1 + (\tau_2 - 1)d + \cdots + (\tau_{p-1} - 1)d^{p-2} \leq r\}$. Let $\mathcal{T} \in \mathbb{R}^{d \times \cdots d}$ be the $p$th order tensor defined by

$$\mathcal{T}_{i_1, i_2, \cdots, i_p} = \sum_{\tau_1 = 1}^{d} \sum_{\tau_2 = 1}^{d} \cdots \sum_{\tau_{p-1} = 1}^{d} \mathcal{A}^{(1)}_{i_1, \tau_1, \tau_2, \cdots, \tau_{p-1}} \mathcal{A}^{(2)}_{i_2, \tau_1, \tau_2, \cdots, \tau_{p-1}} \cdots \mathcal{A}^{(p)}_{i_p, \tau_1, \tau_2, \cdots, \tau_{p-1}}$$

for all $i_1, \cdots, i_p \in [d]$. It can easily be checked that $\mathcal{T}$ is a tensor of CP rank at most $r$, i.e., $\mathcal{T} \in \mathcal{T}(G_{\mathrm{CP}}(r))$. Indeed, from the definition of $\mathcal{A}^{(1)}$, we have

$$\mathcal{T}_{i_1, i_2, \cdots, i_p} = \sum_{\tau_1 = 1}^{d} \sum_{\tau_2 = 1}^{d} \cdots \sum_{\tau_{p-1} = 1}^{d} \mathcal{A}^{(1)}_{i_1, \tau_1, \tau_2, \cdots, \tau_{p-1}} \mathcal{A}^{(2)}_{i_2, \tau_1, \tau_2, \cdots, \tau_{p-1}} \cdots \mathcal{A}^{(p)}_{i_p, \tau_1, \tau_2, \cdots, \tau_{p-1}}$$

$$= \sum_{(\tau_1, \cdots, \tau_{p-1}) \in S} \mathcal{A}^{(1)}_{i_1, \tau_1, \tau_2, \cdots, \tau_{p-1}} \mathcal{A}^{(2)}_{i_2, \tau_1, \tau_2, \cdots, \tau_{p-1}} \cdots \mathcal{A}^{(p)}_{i_p, \tau_1, \tau_2, \cdots, \tau_{p-1}}$$

where the sum is over at most $r$ terms (from the definition of $S$). At the same time, we have

$$\mathcal{T}_{i_1, i_2, \cdots, i_p} = \sum_{(\tau_1, \cdots, \tau_{p-1}) \in S} \mathcal{A}^{(1)}_{i_1, \tau_1, \tau_2, \cdots, \tau_{p-1}} \mathcal{A}^{(2)}_{i_2, \tau_1, \tau_2, \cdots, \tau_{p-1}} \cdots \mathcal{A}^{(p)}_{i_p, \tau_1, \tau_2, \cdots, \tau_{p-1}}$$

$$= \sum_{(\tau_1, \cdots, \tau_{p-1}) \in S} \mathcal{A}^{(1)}_{i_1, \tau_1, \tau_2, \cdots, \tau_{p-1}} \delta_{i_2, \tau_1} \delta_{i_3, \tau_2} \cdots \delta_{i_p, \tau_{p-1}}$$

$$= \begin{cases} \mathbf{A}_{i_1, i_2 + (i_3 - 1)d + \cdots + (i_p - 1)d^{p-2}} & \text{if } i_2 + (i_3 - 1)d + \cdots + (i_p - 1)d^{p-2} \leq r \\ 0 & \text{otherwise} \end{cases}$$

Hence, each one of the components of $\mathbf{A}$ appears exactly once in $\mathcal{T}$. In particular, this implies that the set of indices

$$\{(i_1, \cdots, i_p) \in [d] \times \cdots \times [d] \mid i_2 + (i_3 - 1)d + \cdots + (i_p - 1)d^{p-2} \leq r\}$$

of size $rd$ is shattered by $\mathcal{T}(G_{\mathrm{CP}}(r))$. The theorem then directly follows from Lemma 10. $\qquad \square$

# B  Experiments

To evaluate the theoretical upper bound provided in Theorem 4, we perform a simple binary classification experiment with synthetic data. We draw a random low rank TT target tensor $\mathcal{W} \in \mathbb{R}^{4 \times 4 \times 4 \times 4}$ of rank 8 by drawing the components of the cores of the TT decomposition i.i.d. from a uniform distribution between -1 and 1. Input-output data is generated with $y_i = \mathrm{sign}(\langle \mathcal{W}, \mathcal{X}_i \rangle)$ for training and testing, where the components of $\mathcal{X}_i$ are drawn i.i.d. from a normal distribution. Using the cross-entropy as loss function, we optimize the empirical risk using stochastic gradient descent with a learning rate of $10^{-2}$ to learn a TT hypothesis of rank $r$.

In Figure 3, we report the log generalization gap of the learned hypothesis $h$, $\log(R(h) - \hat{R}_S(h))$, where the true risk $R(h)$ is estimated on a test set of size $4,000$ for different scenarios. In Figure 3 (left), we show how the sample size affects the generalization gap for learned hypothesis of rank $r = 2$ and $r = 4$. As expected, the generalization gap decreases as the sample size grows, and is smaller for $r = 2$ than $r = 4$ which is also expected from the Theorem 4. In Figure 3 (right), we show how the rank $r$ of the learned hypothesis affects the generalization for samples sizes 2,000 and 4,000. As expected, the higher the rank of the TT weight tensor, the larger the model complexity and hence the generalization gap. In both figures, we observe that the theoretical upper bound and the experimental results follow a similar trend as a function of sample size and hypothesis rank.

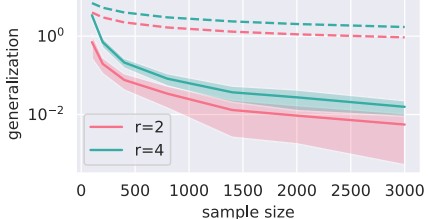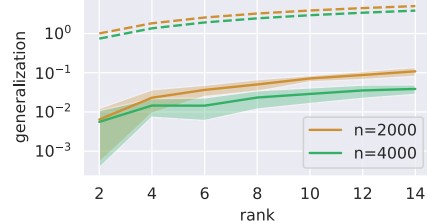

Figure 3: Dashed lines represent the theoretical bound, full lines represent the log generalization gap (averaged over 20 runs for both experiments), and shaded areas show the standard deviation. (left) Generalization error for two models with ranks $r = 2$ and $r = 4$ as a function of training size. (right) Generalization error for two sample sizes $n = 2000$ and $n = 4000$ as a function of the rank of the learned hypothesis.