# OpenReview forum: "Lower and Upper Bounds on the Pseudo-Dimension of Tensor Network Models"
_NeurIPS.cc/2021/Conference — NeurIPS 2021 Spotlight_

### Official Review · Reviewer_sBQh · 2021-07-16

**Rating:** 7
**Confidence:** 3

**Summary:**

This paper studies the Pseudo-dimension (also the VC-dimension for the classification task) of a class of learning models (including linear regression (Eq. 4) and classification (Eq. 5), and completion (Eq. 6) ), of which the weights are modeled by tensor networks (TNs). In the work, the authors give an upper bound on the Pseduo-dimension of the models (Thm. 2) and a generalization bound for classifiers (Thm. 4). Furthermore, the authors also discuss the lower-bounds for several specific TN models to demonstrate the tightness of the obtained upper-bounds up to a $\mathcal{O}(\log(p))$ factor.

**Limitations And Societal Impact:**

Yes, the authors adequately addressed the limitations of the work.

**Main Review:**

**Pros:**

- The proposed results are important for researchers to understand how different TNs impact the learning capacity of the TN model.
- In several cases, the new bounds are tighter than the one in the existing works.
- The proof on Thm. 2  is simple yet smart, which would inspire the tensor researchers to explore more results on tensor learning  along this line in the future.

### Some specific comments and questions:

1. As illustrated by Thm. 2, the Pdim or dvc are bounded by $2N_G\log(12\vert{}V\vert)$, where $N_G$ denotes the total number of the TN and $\vert{}V\vert$ corresponds to the order (or degree) of TN to model the weight tensor. Given a fixed $N_G$, may I say that a higher-order tensorization (i.e., a larger $\vert{}V\vert$) can generally (or always) result in better learning capacity of the model, if we ignore the training issues for higher-order tensor models?
2. As shown in Thm. 3, the claim holds if $n>N>2$. Does it mean that the main results in the work cannot be used to interpret those over-parameterized models, which are popularly applied in the settings of deep learning?
3. The results in the work seem to imply that the TN structures are not so important for the learning capacity, as long as the scale of $N_G$ and $\vert{}V\vert$ are guaranteed. This claim would impact the study on TN structure search if this inference is true.
4. I'm curious if the proposed results can also be used to interpret a similar bound given in the following work?

> *Li, Jingling, et al. "Understanding generalization in deep learning via tensor methods." International Conference on Artificial Intelligence and Statistics. PMLR, 2020.*

**Time Spent Reviewing:**

5-6

---

> ### Author Response · Authors · 2021-08-10
> **Answer to Reviewer sBQh**
>
> We thank the reviewer for their time and careful review.
>
> > As illustrated by Thm. 2, the Pdim or VC-dim are bounded by $2(N_G)log⁡(12|V|)$, where $N_G$ denotes the total number of the TN parameters and |V| corresponds to the order (or degree) of TN to model the weight tensor. Given a fixed $N_G$, may I say that a higher-order tensorization (i.e., a larger |V|) can generally (or always) result in better learning capacity of the model, if we ignore the training issues for higher-order tensor models?
>
> Yes, that is a very good point: assuming our upper bound is tight, for a fixed number of parameters $N_G$, the VC dimension will grow with the number of vertices in the TN, thus a higher-order tensorization increases the capacity (as measured by the VC dimension). Thanks for pointing this out, we will comment on this point in the revision.
>
>
> > As shown in Thm. 3, the claim holds if n>N>2. Does it mean that the main results in the work cannot be used to interpret those over-parameterized models, which are popularly applied in the settings of deep learning?
>
> No, our main results are not affected by this assumption of Theorem 3: the VC dimension is well-defined for any class of hypotheses, independently of any specific dataset or learning regime. It is thus a relevant notion also for neural networks and other models, independent of the “learning regime” that is usually used for such models, either over-parameterized or under-parameterized. A recent example is:
>
> Bartlett PL, Harvey N, Liaw C, Mehrabian A. Nearly-tight VC-dimension and pseudodimension bounds for piecewise linear neural networks. The Journal of Machine Learning Research. 2019 Jan 1;20(1):2285-301
>
> where the authors study the VC dimension of deep neural networks with RELU activation functions. Notice that to derive their upper bound, they also use a variant of Warren’s theorem with the same constraint $n>N$ (Lemma 17 in their work).
>
> > The results in the work seem to imply that the TN structures are not so important for the learning capacity, as long as the scale of $N_G$ and $|V|$ are guaranteed. This claim would impact the study on TN structure search if this inference is true.
>
> This is indeed true for the VC dimension of TN models (assuming our upper bounds are tight, which is the case for e.g., TT and TR up to the logarithmic term). In particular, the hypothesis classes corresponding to TT and TR have the same VC dimension even though the network topologies are significantly different (acyclic vs. cyclic). However, notice that the VC-dimension is one (combinatorial) measure of complexity out of many, and other complexity measures could have more dependence on the TN structures. Also, note that even though the TN structure does not impact the VC dimension of the corresponding classes, it is known that the TN topology can significantly impact the optimization process (e.g., loopy TN are known to be more ‘difficult’ to optimize).
>
>
> >I'm curious if the proposed results can also be used to interpret a similar bound given in the following work?
> Li, Jingling, et al. "Understanding generalization in deep learning via tensor methods." International Conference on Artificial Intelligence and Statistics. PMLR, 2020.
>
> We believe that our results cannot  be used to find a similar bound to theirs. That is mainly because of the difference between the models considered in these two works. While we are dealing with linear TN models (with arbitrary TN structures), Li, Jingling et al. study fully connected neural networks with layers compressed using the CP decomposition. To make meaningful connections between the two studies, a first step could be to extend their work to include other TN structures as layer compressors, or to extend our work to non-linear TN-models. This is an interesting future research direction which we will mention in the revision.

---

### Official Review · Reviewer_rJuG · 2021-07-16

**Rating:** 7
**Confidence:** 4

**Summary:**

The paper presents generalization bounds for arbitrary Tensor-based network learning models. By carefully defining the space of functions that can be defined by certain tensor decompositions, they upper bound and lower bound the VC and pseudo-dimension of direct regressors, classifiers, and completion setups, leading to simple generalization bounds.

**Limitations And Societal Impact:**

I agree with the authors that the theoretical analysis presented here does not warrant any discussion of potentially negative societal impact. Limitations and future work are adequately discussed in the conclusion.

**Main Review:**

Clarity:
a) The paper is extremely well written and easy to follow. The authors do a fantastic job in easing the reader through all relevant background and motivation, as well as clearly and simply explaining technical detail important for understanding various steps in their work. The paper is a pleasure to read.

Quality:
b) The results presented are both simple and interesting. While Tensor-based models have been more and more frequently used in machine learning pipelines, the analysis of these structures has been limited to simply describing the capacity of TNs and drawing equivalences to existing machine learning models (RNNs to TTs, CNNs to Hierarchical Tucker, etc). The results here provide a very simple foundation on which more standard learning theoretic analysis can be defined over tensor-based networks.

c) I would have liked to see some more significant connections drawn with existing work on “expressive power” of networks ([13,30,16,2]), particularly on connecting their definitions of capacity to generalization as described here.

Novelty:
d) The theorems and bounds are somewhat simple, and while they do take advantage of specific tensor network constructions, end up typically reducing to a bounds loosely similar to an unconstrained space over the compressed number of parameters up to some log factor.

Other:
Literature review seems fairly complete.
I have looked briefly at the supplementary material and follow general proof directions but have not vetted the theoretical analysis thoroughly.
Supplementary notebook allows for replication of basic experiment conduced in supplement.

Final Thoughts:
An argument could be made that the results are too basic (low novelty), but in my mind having this as a starting point for accessing generalization with tensor networks will be a valuable step in future theoretical analysis of TNs, as described by the authors in their conclusion.


**Time Spent Reviewing:**

3

---

> ### Author Response · Authors · 2021-08-10
> **Answer to Reviewer rJuG**
>
> We thank the reviewer for their careful review and constructive feedback. We discuss one particular point raised in the review:
>
> > I would have liked to see some more significant connections drawn with existing work on “expressive power” of networks ([13,30,16,2]), particularly on connecting their definitions of capacity to generalization as described here.
>
> This is a very important and interesting point indeed. While [13,30,16,2] are indeed very relevant to our work, we first want to clarify some differences:
> - [2,16] focus on unsupervised learning while our analysis is focused on the supervised case.
> - All these papers focus on particular TN structures (Hierarchical Tucker, CP and TT for [13,30] and “TT-like” TN structures for [16,2]), while our results and analysis hold for arbitrary TN structures.
> - Lastly, in order to make an analogy with Convolutional Arithmetic Circuits (CAC), [13] considers the special case where the input tensor is rank one. In contrast, our analysis does not make any assumption on the input tensors (and thus holds for the special case considered in [13] and [54] in addition to more general settings).
>
> Nonetheless, we agree that drawing such connections is a very exciting direction for future research (though not trivial in our opinion). In particular, we think [13] is the most relevant work and we believe our results are complementary to theirs and differ by the way the capacity is measured: in [13], the authors study to which extent a hierarchical Tucker tensor can be approximated arbitrarily well by a CP tensor, whereas we study the VC-dimension. To some extent, this implies that the results in [13] are about expressiveness (outside of any learning paradigm) whereas our results are closer to the learning setting (e.g., generalization bounds can be derived from the VC dimension). Another interesting point to mention is that one cannot derive bounds on the VC-dimension of the TN models from the main results of [13]; that is, our results are not consequences of the ones in [13], nor the other way around. But again, we agree that drawing more significant connections with [13] is a very interesting direction that we are currently investigating (e.g., using our analysis to derive results on the generalization abilities of various CAC architectures). We will mention this relevant future direction in the revision.

---

### Official Review · Reviewer_1sdU · 2021-07-16

**Rating:** 6
**Confidence:** 3

**Summary:**

This work considers the tensor network methods in machine learning. The major contributions are some new lower and upper bounds on the pseudo-dimension for a large class of TN models. The derived results can be applied to machine learning tasks which involves the tensor decomposition models.

**Limitations And Societal Impact:**

See Cons above.

**Main Review:**

Pros:
This work is well organized, the writing is clear and easy to follow.

The results are valid, there are proofs for each Theorem and Corollary stated in the paper.

Cons:
It is mentioned that the derived bound can be used in many machine learning tasks like matrix completion, regression, and classification problems. However, how these bounds can be used in these tasks and how these bounds can help improve performance or facilitate the analyses are not clear. It would be better that some discussions about this are included.

It would be better that some simple simulations like matrix completion were included to show the effect of the proposed method.

**Time Spent Reviewing:**

2h

---

> ### Author Response · Authors · 2021-08-10
> **Answer to Reviewer 1sdU**
>
> We thank the reviewer for their time and feedback.
>
> > It is mentioned that the derived bound can be used in many machine learning tasks like matrix completion, regression, and classification problems. However, how these bounds can be used in these tasks and how these bounds can help improve performance or facilitate the analyses are not clear. It would be better that some discussions about this are included.
>
> Our work provides a first step towards understanding the generalization ability of TN methods and designing principled algorithms for TN in ML. A first interesting direction building upon our results would be to design structural risk minimization approaches for TN models. We also believe that the technical tools we used to derive our results will facilitate future analyses of TN models (e.g., showing how Warren's theorem can be used to characterize the expressiveness of TN models).
>
> > It would be better that some simple simulations like matrix completion were included to show the effect of the proposed method.
>
> Note that, at this point, our work is mainly theoretical and we do not propose a method or algorithm. We also included a simple classification experiment in the supplementary material where we compare the generalization error predicted from our upper bound with the one observed in practice.

---

### Decision · Program_Chairs · 2021-09-27

**Decision:**

Accept (Spotlight)

**Comment:**

The reviewers clearly appreciated the paper, and asked a number of questions that triggered interesting exchanges with the authors, as well as a couple of useful clarifications.
The authors are encouraged to take into account these exchanges when preparing the final version of their manuscript.